# The role of carbon nanoparticle in lymph node detection and parathyroid gland protection during thyroidectomy for non-anaplastic thyroid carcinoma- a meta-analysis

**Shaowei Xu[1], Zhifeng Li[2], Manbin Xu[1], Hanwei Peng[1]***

**1** Department of Head and Neck Surgery, Cancer Hospital, Shantou University Medical College, Shantou, China, **2** Department of Oncology, Cancer Hospital, Shantou University Medical College, Shantou, China

* penghanwei@126.com

**Data Availability Statement:** All relevant data are within the paper and its Supporting Information files.

## Abstract

### Objective

To assess the efficiency of the carbon nanoparticles (CNs) in lymph node identification and parathyroid gland (PG) protection during thyroidectomy for non-anaplastic thyroid carcinoma (N-ATC).

### Methods

A systematic literature search for relevant literatures published up to December 2018 in PubMed, EMBASE, Web of Science and Cochrane Library was performed. Both English and Chinese literatures were retrieved and analyzed. Randomized controlled trials or non-randomized controlled trials comparing the use of CNs with the use of methylene blue or a blank control in patients undergoing thyroidectomy for N-ATC were enrolled in this study. The primary outcomes included the number of lymph nodes harvested, the rate of lymph nodes involved, and the rates of accidental parathyroidectomy, hypoparathyroidism, and hypocalcemia. Weighted mean differences (WMDs), odds ratios (ORs) and risk differences (RDs) were calculated for the dichotomous outcome variables. Between study heterogeneity was tested using the $Q$ tests and the $I^2$ statistics. All analyses were performed using Review Manager (version 5.3.5).

### Results

25 studies comprising 3266 patients were included in this analysis. The total number of lymph nodes harvested in the CNs groups was significantly higher than that in the control groups (WMD, 2.36; 95% CI, 1.40 to 3.32; $P$ <0.01). Administrating CNs was associated with a lower incidence of accidental PG removal (OR = 0.28, 95% CI = 0.21 to 0.37, $P$<0.01) and lower rates of both postoperative transient hypoparathyroidism (OR = 0.46, 95% CI = 0.33 to 0.64, $P$ <0.01) and transient hypocalcemia (OR = 0.46, 95% CI = 0.33 to 0.65, $P$ <0.01). No significant difference was found concerning lymph node metastatic rates between CNs group and control group. Subgroup analysis indicated that the application of CNs in reoperation thyroidectomy reduced both the rate of transient hypoparathyroidism

**Funding:** Shaowei Xu Grant numberts: 2018A004 Shantou University Medical College Cancer Hospital http://www.sumcch.cn/. The funders had no role in study design, data collection and analysis, decision to publish, or preparation of the manuscript.

**Competing interests:** The authors have declared that no competing interests exist.

(OR = 0.21, 95% CI = 0.06 to 0.75, $P$ = 0.02) and the possibility of accidental PGs removal (OR = 0.21, 95% CI = 0.07 to 0.62, $P$ = 0.004, $P$<0.05).

## Conclusions

The application of CNs in thyroidectomy for N-ATC results in higher number of lymph node harvested and better PG protection during both initial and reoperation thyroidectomy.

## Introduction

Thyroid cancer is one of the most common types of cancer in the world, the incidence of which increased dramatically in the recent two decades [1]. More than 95 percent of the thyroid carcinomas were non-anaplastic thyroid carcinoma (N-ATC), including papillary thyroid carcinoma, follicular thyroid carcinoma, and medullary thyroid carcinoma [1]. Total thyroidectomy combined with level II-VI neck dissection for clinically N1b cases or central compartmental dissection (CND) for cN0 cases is widely advocated as one of the standard treatment protocols for N-ATC, whereas lobectomy is only accepted for T1 cases without high risk factors [2–4]. However, it's a challenge for surgeons to perform a total thyroidectomy with or without CND due to the potential risk of postoperative transient hypocalcemia/hypoparathyroidism (incidence rate 20%-60%) or permanent ones (incidence rate 2%-7%) based on the previous reports and even higher for those who undergo a reoperation [5, 6].

Carbon nanoparticles (CNs) have been successfully attempted for lymph node dissection in breast carcinoma, gastric carcinoma, and a few other malignancies [7, 8]. CNs have a mean diameter of 150 nm larger than the size of the capillary endothelial cell gap (20–50 nm), and thus cannot enter the blood vessels. However, they can penetrate the lymphatic capillary endothelial cell gap (120–500 nm) and may be phagocytized by macrophages. Thus, these CNs specifically accumulate in the lymph nodes, staining them black and easily been identified with naked eyes [9, 10]. However, the parathyroid glands (PG) remains unstained due to their possible lack of lymphatics. This so called "negative development" not only increases the number of lymph nodes harvested during compartmental dissection, but also facilitates the surgeons' distinguishing of the PGs from the lymph nodes and decreases the possibility of unintentional removal of PGs [11–13]. The pattern of CNs metabolism is the hypothetic basis that CNs can be used as a tracer to detect the sentinel nodes.

In the past decade, CNs had been successfully attempted as a negative developer to protect parathyroid gland (PG) during initial thyroidectomy [14]. Although a couple of meta-analyses evaluating the value of CNs in initial thyroidectomy had been published, the data need to be updated [15, 16]. Furthermore, there are still doubts about the efficiency of CNs in reoperation thyroidectomy due to the hypothesis that the lymphatic capillaries may be destroyed during initial surgery [17, 18]. Therefore, we performed a meta-analysis with more comprehensive and updated studies to summarize the role of CNs in lymph node detection and PG protection during both initial and reoperation thyroidectomy.

## Materials and methods

### Search strategy

Two authors conducted independently a search for relevant literatures up to December 2018 in PubMed, EMBASE, Web of Science, and Cochrane Library. The following medical search headings were used: ("thyroid gland"[MeSH Terms] OR ("thyroid"[All Fields] AND

"gland"[All Fields]) OR "thyroid gland"[All Fields] OR "thyroid"[All Fields] OR "thyroid (usp)"[MeSH Terms] OR ("thyroid"[All Fields] AND "(usp)"[All Fields]) OR "thyroid (usp)"[All Fields]) and ("carbon"[MeSH Terms] OR "carbon"[All Fields]). We manually searched the references of eligible studies and ClinicalTrials.gov to ensure identification of relevant published and unpublished studies. Chance-corrected agreement (i.e. kappa) in the screening stages and the eventual final set of included studies was calculated.

### Inclusion criteria

Studies included in the meta-analysis need to fulfill the following criteria: (1) human N-ATC confirmed by pathology; (2) patients underwent thyroidectomy/lobectomy and/or neck dissection; (3) studies designed to compare the use of CNs with the use of methylene blue (MB) or with blank control; (4) studies on human beings; (5) full text available in English or Chinese.

### Exclusion criteria

Studies were excluded if they (1) had a sample size less than 15; (2) included pregnancy or adolescent (aged<16); (3) included patients with benign and malignant thyroid diseases, and complete data of the malignancies were unavailable. Exclusion criteria (2) was set to reduce the publication bias because most of the studies reported only the cases in adult or adolescents over 16 years old. When two or more studies were reported by the same authors and/or institution, either the most recent study or the higher quality study was included in the analysis to excluded the possible duplicate cases.

### Data extraction

Two reviewers (SW Xu and ZF Li) independently performed the first-stage screening of titles and abstracts based on the inclusion criteria. In the second-stage screening, the two reviewers retrieved and reviewed the possible relevant articles in full text to confirm the included articles, and then they recorded the following data independently: first author, publication year, sample size, description of study population (age, sex), study design (RCT or NRCT), surgical procedure (CNs injection dose, points and waiting time), lymph nodes details (number of harvested and involved LN), parathyroid protection outcome. Outcome of parathyroid protection was evaluated based on the following parameters: number of the PGs identified and/or incidentally removed and patient numbers with transient or permanent postoperative hypoparathyroidism/hypocalcemia. Both hypoparathyroidism and hypocalcemia were accepted as parameters to judge parathyroid function impair, and the threshold values in each report were adopted. When hypoparathyroidism or hypocalcemia persisted for 6 months or more, it was defined as permanent, otherwise it was defined as transient. Any discrepancies were resolved by discussion or referred to the corresponding author (HW Peng).

### Quality assessment

Concerning the quality of study design, the RCT was assessed according to the Jadad Scoring system, which consists of 3 items: randomization (0–2 points), blinding (0–2 points), and descriptions of the withdrawals and dropouts (0 or 1 point). The total possible score was 5 points. The Newcastle-Ottawa Scale was used for NRCT.

### Statistical analysis

All analyses in the current meta-analysis were performed using RevMan 5.3.5 (free software downloaded from http://www.cochrane.org). The results are presented as weighted mean

differences (WMDs) and Odds ratios (ORs) with a 95% confidence interval (CI). A $P$ value< 0.05 was considered statistically significant, except where otherwise specified. Moreover, study heterogeneities were quantified using the $Q$-test and the $I^2$ statistic. When $P > 0.1$ and $I^2 < 50\%$, a fixed-effect model was used; otherwise, a random-effects model was applied. Possible publication bias was tested by Begg's funnel plot.

## Results

### Study selection and description

Fig 1 details the study flowchart of the initial search and the subsequent selection of relevant articles. The initial search retuned 4488 studies. In the first-stage screening, 4202 irrelevant references and 230 duplicates were excluded. The remaining 56 studies with full text were further evaluated in the second-stage screening, 31 studies of which were excluded due to the reasons listed in the diagram. Finally, 25 studies fulfilled the inclusion criteria for the meta-analysis [9–13, 18–37]. Begg's test for 25 studies present in Fig 2 and the Kappa Coefficient was 0.68 (95% CI:0.66–0.70). Table 1 summarizes all the individual studies and their characteristics. Of these 25 studies, 13 were RCT, 12 were NRCT. A total of 3266 patients were included in this meta-analysis, of which 1496 were included in the CN group, 1770 in the Control group (1496 in blank control and 181 in MB control). 3248 patients had PTC; 12 had FTC, and 6 had MTC. The two groups had no significant difference in terms of age (MD, -0.43; 95% CI, -1.35–1.26, $P = 0.95$) and sex (OR, 1.05; 95% CI, 0.87–1.25, $P = 0.63$). There were 3096 initial surgeries and 170 reoperations in this analysis.

The quality assessment details for the RCTs and NRCTs are presented in Tables 1 and 2 in S1 File.

### Surgical procedure

The CNs were provided by Chongqing LUMMY Pharmaceutical Co., Ltd. In most of the studies, the CNs were injected underneath the fibrous thyroid capsules at two or three points around the tumor, 0.1–0.2ml for each point, with 5-10mins wait before thyroidectomy (Table 3 in S1 File).

### Lymph node removal

Analysis of the number of harvested lymph nodes were possible in 21 studies. A random-effects model was applied in this analysis ($P < 0.01$, $I^2 = 97\%$). WMD analysis showed that the total number of harvested lymph nodes in the CN group was significantly higher than that in the control groups (WMD, 2.36; 95% CI, 1.40 to 3.32; $P < 0.01$, Fig 3). The rate of LN black-stained varied between 73.3% and 95.3%. Eighteen out of the 21 studies provided adequate information for the analysis of LN metastatic rate (= number of tumors involved LN/ number of harvested LNs). A random-effects model was applied to assess heterogeneity ($P < 0.01$, $I^2 = 96\%$). No difference was found between the CNs group and the Control group regarding LN metastatic rate (OR = 1.07, 95% CI = 0.75 to 1.51, $P = 0.71$, Fig 4).

### Parathyroid gland protection

A fixed-effects model was used to analyze the data ($P = 0.89$, $I^2 = 0\%$). After analyzing the accidental removal of the PGs using 23 eligible studies, we found that the possibility of accidental PGs removal was decreased by 30% in CNs group compared with control group (OR = 0.28, 95% CI = 0.21 to 0.37, $P < 0.01$, Fig 5).

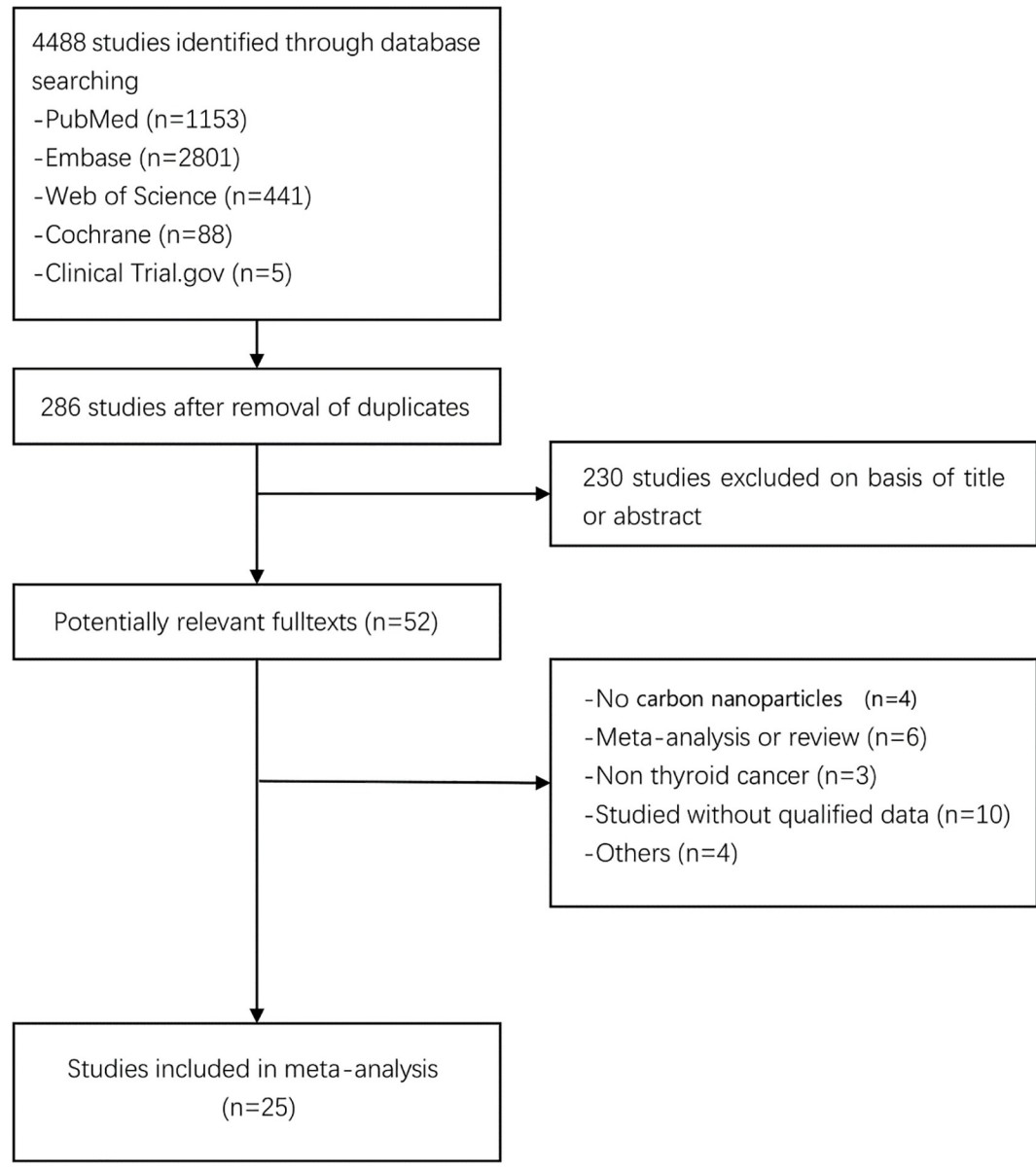

**Fig 1. Study flow diagram.**

The overall postoperative transient hypoparathyroidism data were available in 16 studies. A random-effects model was used, $P<0.04$, $I^2 = 42\%$), and subgroup analysis of transient hypocalcemia was possible 15 studies (A random-effects model was applied, $P<0.08$, $I^2 = 36\%$). Data analysis showed that the application of CN decreased the rate of both postoperative transient hypoparathyroidism and transient hypocalcemia equally by 46% (OR = 0.46, 95% CI = 0.33 to 0.64, $P<0.01$, Table 2, Fig 1 in S2 File; OR = 0.46, 95% CI = 0.33 to 0.65, $P<0.01$, Table 2, Fig 2 in S2 File). A random-effects model was applied to our data ($P = 0.44$, $I^2 = 0\%$), and 7 studies had a followed-up time of at least 6 months, thus the data of permanent hypocalcemia were analyzed. No significant difference concerning permanent hypocalcemia was found between the CNs group and control group (OR = 0.55, 95% CI = 0.09 to 3.43, $P = 0.52$, Table 2, Fig 3 in S2 File).

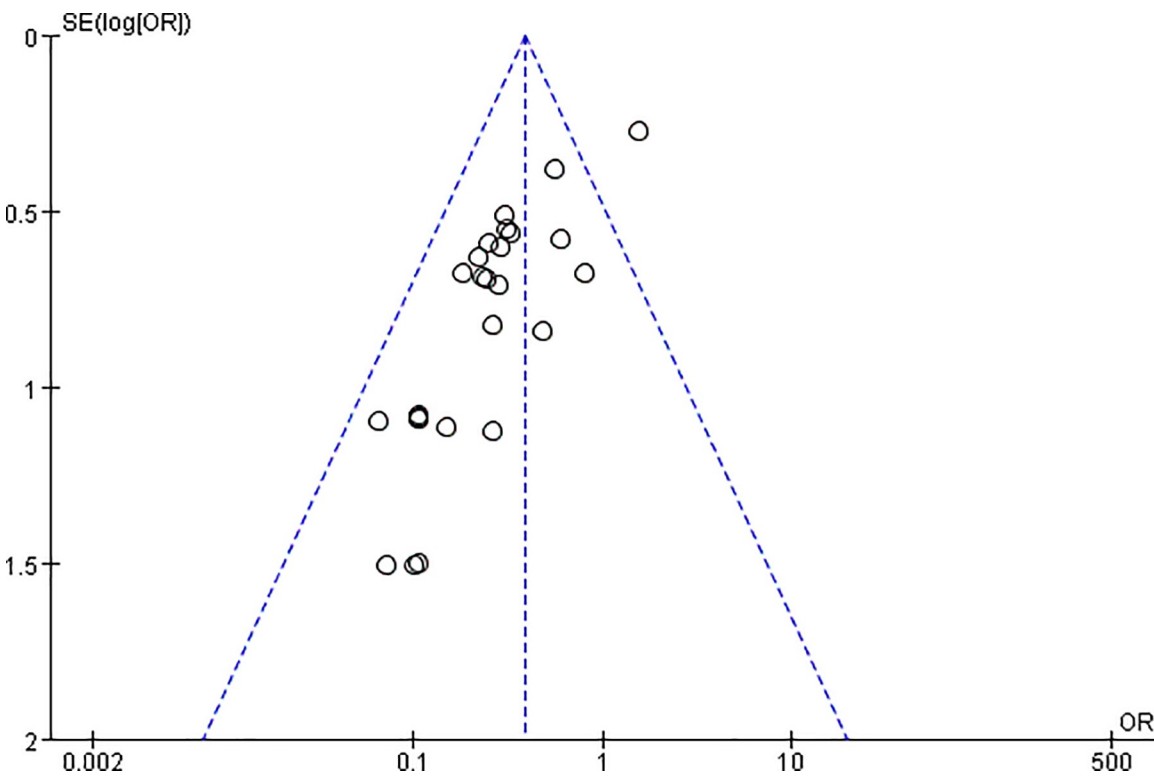

**Fig 2. The funnel plots of the publication bias calculated from the Begg's test.**

### CNs application during reoperation

Two out the 25 included studies focused on reoperation thyroidectomy. Subgroup analysis of the data using a random-effects model demonstrated that the use of CNs significantly reduced the postoperative transient hypoparathyroidism rate by 21% (OR = 0.21, 95% CI = 0.06 to 0.75, *P* = 0.02, Table 2, Fig 4 in S2 File), and the possibility of accidental PGs removal was decreased by 21% (OR = 0.21, 95% CI = 0.07 to 0.62 *P* = 0.004, Table 2, Fig 5 in S2 File).

## Discussion

Protection of the PGs and the recurrent laryngeal nerves are important techniques related to thyroidectomy, particularly total thyroidectomy combined with CND or reoperation thyroidectomy. Anatomically, the location of PGs, particularly the lower two, are variable, which results in a high risk of being accidentally removed during surgery [38]. Removal of all PGs inevitably leads to transient or permanent hypoparathyroidism and hypocalcemia, which has an adverse impact on the quality of life of the patients. Exposure and identification of the PGs intraoperatively facilitates preservation of the PG function. In addition, PTC has a trend of involving youngers and the 5-year survival rate is reported up to 98% [1]. As a result, omission of micro-metastases of the central compartmental lymph nodes may lead to reoperation, which may subsequently increase the incidence of complications and mortality. Therefore, effective techniques are urgently needed to improve the thoroughness of CND while preserving the PGs, particularly for reoperation.

CNs, approved by China Food and Drug Administration, have been efficiently attempted to assist lymph node dissection for gastric cancer, breast cancer and other malignancies [7, 8]. In the recent years, CNs are used as a novel and safe surgical technique to trace the regional

**Table 1. Characteristics of the articles included in the meta-analysis.**

| Study | Study period | Group | Patients (N) | Age | M | F | Etiology |
|---|---|---|---|---|---|---|---|
| Bai, Y.C (2013) | Jun 2010 to Mar 2012 | Experimental | 48 | 46.3±9.2 | 9 | 39 | 48PTC |
| | | Control | 73 | N | 7 | 33 | 73PTC |
| Chaojie (2016) | Feb2011 to Feb 2014 | Experimental | 64 | N | 11 | 53 | 64PTC |
| | | Control | 52 | N | 9 | 43 | 52PTC |
| Chen, W (2014) | Jan2013 to Dec 2013 | Experimental | 36 | 38.23±10.67 | 5 | 31 | 36PTC |
| | | Control | 36 | 34.64±8.75 | 8 | 28 | 36PTC |
| Deng, W (2014) | July 2011 to Dec 2013 | Experimental | 18 | N | 4 | 14 | 18PTC |
| | | Control | 33 | N | 12 | 21 | 33PTC |
| Fu, H (2017) | Oct 2015 to May 2016 | Experimental | 75 | 45.41±10.2 | 21 | 54 | 75PTC |
| | | Control | 73 | 45.32±9.74 | 21 | 52 | 7PTC |
| Gao, B (2015) | Jan 2012 to Dec 2014 | Experimental | 27 | 49.4±2.5 | 2 | 25 | 27PTC |
| | | Control | 27 | 52.5±1.8 | 2 | 25 | 27PTC |
| Gao, Q (2014) | Jan2010 to Oct 2012 | Experimental | 50 | 42±1.28 | 12 | 38 | 50PTC |
| | | Control | 50 | N | 9 | 41 | 50PTC |
| Gu, J (2015) | Jun 2012 and Aug 2014 | Experimental | 50 | 46.98±9.027 | 10 | 40 | 47PTC+1FTC+2MTC |
| | | Control | 50 | 47.76±13.91 | 6 | 44 | 48PTC+1FTC+1MTC |
| Hao, R. T (2012) | Jan2008 to Dec 2009 | Experimental | 100 | 41 | 14 | 86 | 100PTC |
| | | Control | 100 | 44 | 11 | 89 | 100PTC |
| Liu, F (2017) | Apr 2016 and Feb 2017 | Experimental | 48 | 51.98 ± 3.1 | N | N | 48PTC |
| | | Control | 48 | 51.98 ± 3.1 | N | N | 48PTC |
| Liu, Y (2018) | Feb 2013 to May 2015 | Experimental | 45 | 46.17 ±10.20 | 17 | 28 | 45PTC |
| | | Control | 47 | 45.39±12.03 | 12 | 35 | 47PTC |
| Long, M (2017) | Jan2012 to May 2013 | Experimental | 42 | 44.5 ± 9.6 | 9 | 33 | 42PTC |
| | | Control | 46 | 43.8 ± 10.3 | 11 | 35 | 46PTC |
| Shen, H (2014) | Mar 2012 to Jul 2013 | Experimental | 45 | N | N | N | 42PTC+3FTC |
| | | Control | 64 | N | N | N | 57PTC+6FTC+1MTC |
| Shi, C (2016) | Jan2014 and Feb 2015 | Experimental | 52 | 45.2±5.8 | 6 | 46 | 52PTC |
| | | Control | 45 | 42 ±4.3 | 6 | 39 | 45PTC |
| Su, A. P (2016) | Apr 2013 to Mar 2015 | Experimental | 195 | 43.38 ± 11.65 | 57 | 138 | 55PTC |
| | | Control | 181 | 45.67 ± 13.50 | 55 | 126 | 181PTC |
| Tian, W (2014) | Apr 2012 to Oct 2013 | Experimental | 50 | 36.4±2.5 | 5 | 45 | 50PTC |
| | | Control | 50 | 44.5±5.8 | 11 | 39 | 46PTC+4FTC |
| Wang, B (2015) | Mar 2013 to Mar 2014 | Experimental | 28 | 30.25 ± 6.04 | 1 | 27 | 28PTC |
| | | Control | 27 | 29.44 ± 6.27 | 2 | 25 | 27PTC |
| Wang, B (2016) | Jan2013 to Jan2014 | Experimental | 90 | 44.36 ± 11.48 | 25 | 65 | 90PTC |
| | | Control | 141 | 44.09 ± 12.41 | 37 | 104 | 141PTC |
| Wang, X. L (2009) | NM | Experimental | 18 | 44 | 10 | 8 | 17PTC+1MTC |
| | | Control | 18 | 40 | 7 | 11 | 16PTC+1FTC+1MTC |
| Xu, X. F (2017) | Sep 2013 to Aug 2014 | Experimental | 57 | 45.37±10.71 | 5 | 52 | 57PTC |
| | | Control | 57 | 42.68±14.43 | 4 | 53 | 57PTC |
| Xue, S (2018) | Jan2010 to Dec 2012 | Experimental | 106 | 44.88 ± 7.78 | 20 | 86 | 106PTC |
| | | Control | 300 | 44.35 ± 10.28 | 66 | 234 | 300PTC |
| Yu, W (2016). | Aug 2012 to Jun 2013 | Experimental | 41 | 41.6±17.1 | 8 | 33 | 41PTC |
| | | Control | 41 | 41.7±18.9 | 11 | 30 | 41PTC |
| Yu, WB (2016) | Jan2012 to Jun 2013 | Experimental | 70 | 44.5 ± 17.4 | 14 | 56 | 70PTC |
| | | Control | 70 | 45.5 ± 19.0 | 17 | 53 | 70PTC |
| Zhu H (2017) | Jan 2013 to Feb 2015 | Experimental | 60 | 60.4±7.18 | 24 | 36 | 60PTC |

*(Continued)*

**Table 1.** (Continued)

| Study | Study period | Group | Patients (N) | Age | M | F | Etiology |
|---|---|---|---|---|---|---|---|
| | | Control | 60 | 62.5±7.65 | 25 | 35 | 60PTC |
| Zhu, Y (2016) | Apr 2010 to Apr 2011 | Experimental | 81 | 46.75±12.09 | 14 | 67 | 81PTC |
| | | Control | 81 | 44.31± 10.37 | 16 | 65 | 81PTC |

M/F = male/female; CN = carbon nanoparticles; PTC = papillary thyroid cancer; FTC = follicular thyroid cancer; MTC = medullary thyroid cancer

lymph nodes and localize the PG during thyroidectomy [23]. So far, no toxic side effects have been reported in humans. However, when the CNs were injected improperly and they spread out of the thyroid capsule, they may stain the surgical field, which makes surgical procedure more difficult [23]. In most of the included studies, the dose of CNs is 0.1 to 0.2 ml per point and 2–3 points around tumor within the thyroid capsule were recommended. After being properly injected, the CNs enter the lymphatic capillaries and stain lymph nodes in 10-15min, rather than PGs and the recurrent laryngeal nerve [10, 13, 18–20, 27, 32–37]. It is hypothesized that this technique facilitates intraoperative identification of LNs and improves thoroughness of LN dissection for thyroid carcinoma.

The current meta-analysis demonstrates that the total number of lymph node harvested in the CN group was approximately 2.36 more than that in the control groups. But the ratio of metastatic lymph nodes has no significant difference between the two groups. The results were similar to previous studies. This denotes that CNs play a key role in accurate identification of the lymph nodes, but they cannot improve the detection of metastatic lymph nodes. The

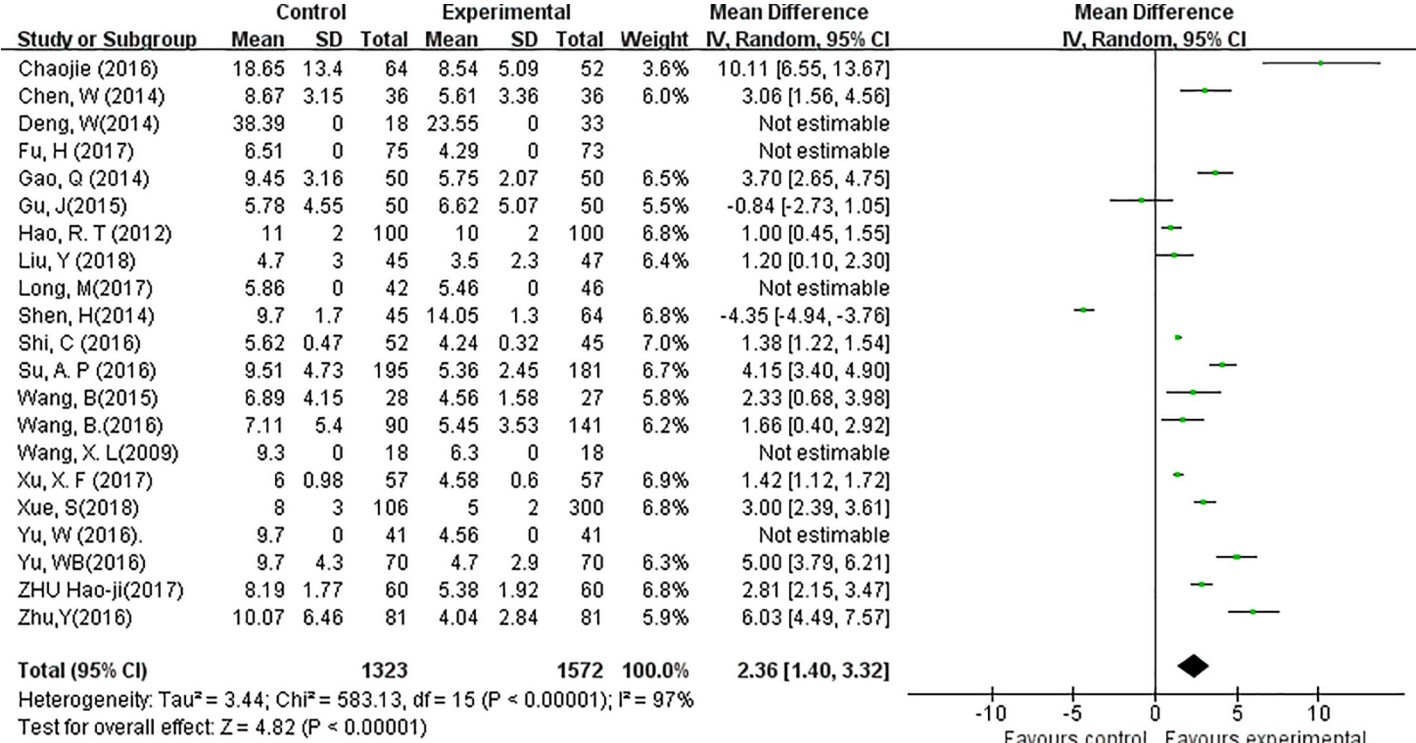

**Fig 3. Forest plots of the total number of lymph nodes harvested in groups.** (Experimental = Carbon nanoparticle group, Control = Blank or methylene blue group, Total = total number of LN harvested, Mean = the average number of LN harvested in each patient, SD = Standard deviation).

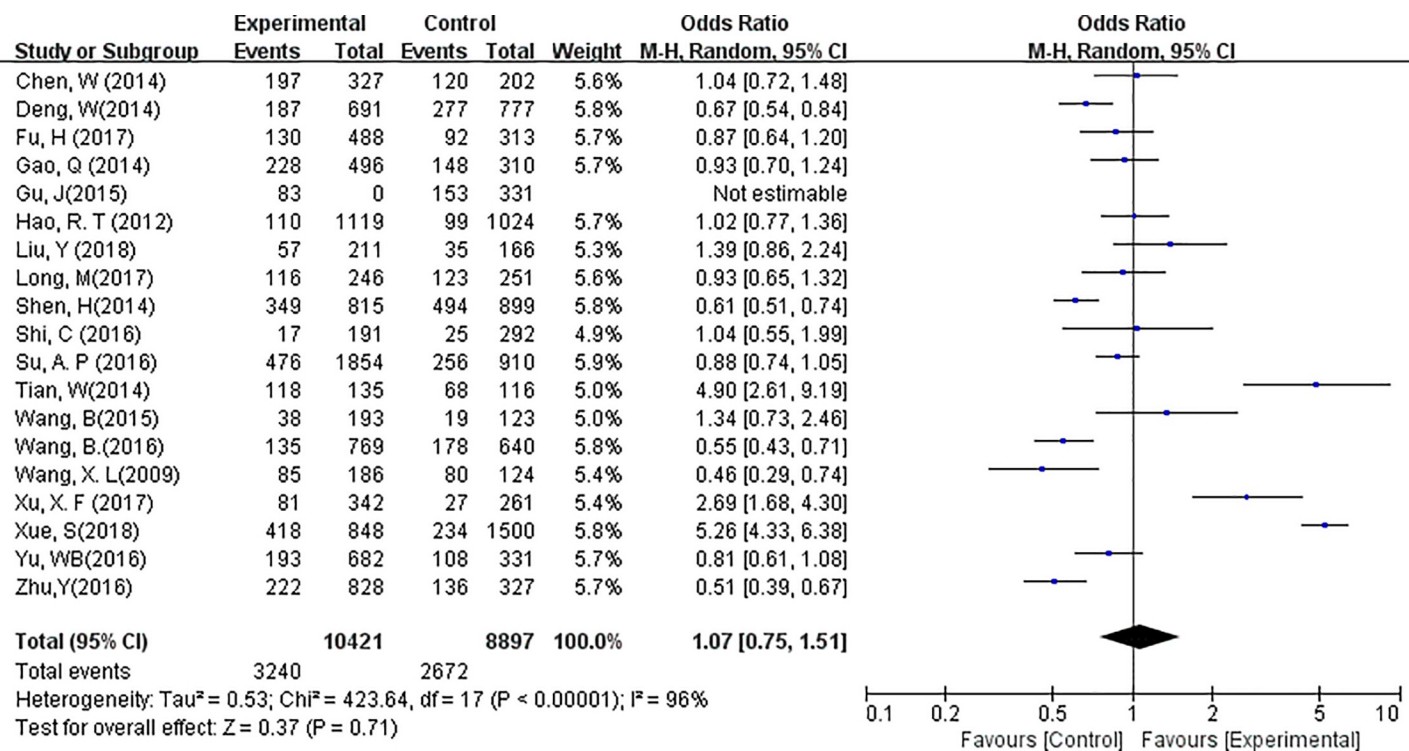

**Fig 4. Forest plots of the total metastatic rate of lymph nodes harvested in groups.** (Experimental = Carbon nanoparticle group, Control = Blank or methylene blue group, Total = total number of LN, Events = the number of LN with metastasis).

possible explanation is that CNs facilitate identification of the tiny LNs which are black dyed by both surgeons and pathologists. However, CNs have no tumor trophism and they dye both the normal LNs and the metastatic LNs black without preference [26, 35].

Hypoparathyroidism is a common complication after thyroid surgery, especially after reoperation.

It is a disorder characterized by hypocalcemia, low or inappropriately normal intact parathyroid hormone (PTH) levels, and often hyperphosphatemia, which may be transient when it recovers within a few weeks or one month after thyroid surgery, or permanent when hypoparathyroidism persists for at least six months postoperatively. The incidence of transient hypoparathyroidism had been reported to be 20% to 60%, while that of permanent hypoparathyroidism was 0% to 7% [5, 6]. Our study revealed that application of CNs during thyroidectomy with or without CND reduces the incidence of accidental PG removal by approximately 30% and reduces the incidence of both postoperative transient hypoparathyroidism and transient hypocalcemia equally by 46%. However, there were still doubts about whether this technique leads to protection of long-term parathyroid function. In our subgroup analysis of permanent hypocalcemia using 7 studies who had a followed-up period of at least 6 months, no significant better PG function outcome was found in CNs group. We hypothesize that CNs facilitate identification of the PGs and lead to better protection of the glands' structures and blood supply. However, the benefit still cannot reach a statistic significance base on the data of these 7 studies yet.

It is assumed that lymphatic tracer cannot accurately drains to the sentinel nodes, the first lymphatic station from the primary cancer in case of reoperation, because the local structures could possibly be destroyed in the initial surgery. Whether previous surgeries disturb the role

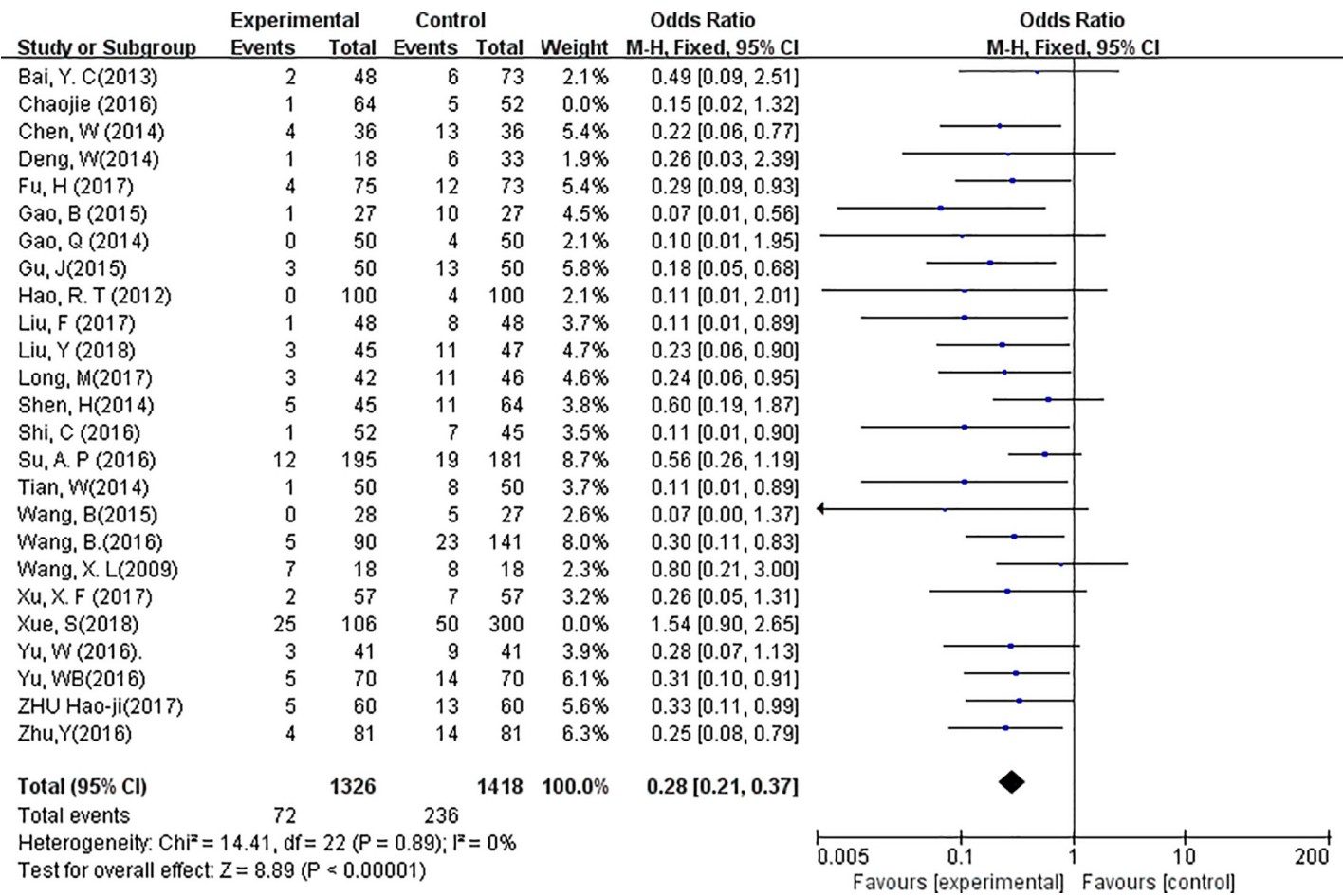

**Fig 5. Forest plots of accidental parathyroid removal rate in groups.** (Experimental = Carbon nanoparticle group, Control = Blank or methylene blue group, Total = The number of patient, Events = the number of parathyroid glands removed accidentally).

of CNs in black-staining the lymph nodes and negative developing the PGs during thyroidectomy remains unknown. To the best of our knowledge, no previous meta-analysis concerned the outcome of the application of CNs for reoperation thyroidectomy has been reported so far. Our systemic review retrieved only 2 studies who focused on CNs application for reoperation thyroidectomy. These reports demonstrated CNs technique not only have a benefit of PGs identification but also decrease the incidence of postoperative transient hypoparathyroidism

**Table 2. Summary for subgroup analysis in the meta-analysis.**

| Subgroup analysis | Studies | Experiment (n) | Control (n) | OR | $I^2$(%) | P |
|---|---|---|---|---|---|---|
| The number of LN harvested | 21 | 1323 | 1572 | (WMD)2.36 | 97 | <0.01 |
| The metastatic rate of LN | 18 | 1208 | 1469 | 1.07 | 96 | 0.71 |
| Accidental parathyroid gland removal | 23 | 1326 | 1418 | 0.3 | 0 | <0.01 |
| Transient hypoparathyroidism. | 16 | 1017 | 1275 | 0.46 | 42 | <0.01 |
| Transient hypocalcemia | 15 | 881 | 1147 | 0.46 | 36 | <0.01 |
| Permanent hypocalcemia | 7 | 590 | 840 | 0.50 | 0 | 0.45 |
| Reoperation transient hypoparathyroidism | 2 | 91 | 79 | 0.21 | 25 | 0.02 |
| Reoperation accidental PGs removal | 2 | 91 | 79 | 0.21 | 0 | 0.004 |

by 20% [26, 35]. We believe although the disturbance of the drainage routes of the CNs leads to failure of sentinel node localization, it does not impact their drainage to the regional LNs, which facilitates identification of the black-stained LN and negative developing of the PGs.

The current meta-analysis has some limitations and the results should be interpreted with caution. First, all the studies including in this meta-analysis were conduct only in China and the subjects were all Chinese. Second, long-term PG function outcome is difficult to figure out because most of the studies had a follow-up period of less than 6 months. Third, average Jadad Score and Newcastle-Ottawa score were only 3 and 7.4 respectively, which indicate the study design was of moderate quality. Thus, further research is needed to verify the conclusions of the current study.

## Conclusions

This is the meta-analysis so far, firstly focused on the application of CNs during both initial thyroidectomy and reoperation thyroidectomy for N-ATC. This meta-analysis demonstrates that the application of CN for thyroidectomy improve the lymph node detection and PG protection not only for initial surgery but also reoperation.

## Supporting information

**S1 Checklist. PRISMA 2009 checklist.**
(DOC)

**S1 File.**
(DOCX)

**S2 File.**
(DOCX)

**S3 File.**
(DOCX)

## Author Contributions

**Conceptualization:** Shaowei Xu, Zhifeng Li.

**Data curation:** Shaowei Xu, Manbin Xu.

**Formal analysis:** Shaowei Xu, Zhifeng Li, Manbin Xu, Hanwei Peng.

**Investigation:** Shaowei Xu, Zhifeng Li, Manbin Xu.

**Methodology:** Shaowei Xu, Manbin Xu.

**Supervision:** Hanwei Peng.

**Writing – original draft:** Shaowei Xu.

**Writing – review & editing:** Shaowei Xu, Hanwei Peng.

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
