## [Decision Letter · Decision Letter 0]

18 Nov 2019

PONE-D-19-26418

The Role of Carbon Nanoparticle in Lymph Node Detection and Parathyroid Gland Protection during Thyroidectomy- a Meta Analysis

PLOS ONE

Dear Prof. Peng,

Thank you for submitting your manuscript to PLOS ONE. After careful consideration, we feel that it has merit but does not fully meet PLOS ONE’s publication criteria as it currently stands. Therefore, we invite you to submit a revised version of the manuscript that addresses the points raised during the review process.

We would appreciate receiving your revised manuscript by Jan 02 2020 11:59PM. To enhance the reproducibility of your results, we recommend that if applicable you deposit your laboratory protocols in protocols.io, where a protocol can be assigned its own identifier (DOI) such that it can be cited independently in the future. For instructions see: http://journals.plos.org/plosone/s/submission-guidelines#loc-laboratory-protocols

We look forward to receiving your revised manuscript.

Kind regards,

Biniam Kidane

Academic Editor

PLOS ONE

Journal Requirements:

2. Please provide the full search strategy and full search terms used for at least one database used in the supplementary information.

In addition, please provide additional details regarding the assessment of publication bias in your systematic review and meta-analysis. Specifically, please provide the funnel plots calculated from the Begg's test and/or the regression plots from the Egger's test.

Thank you for your attention to these requests.

Additional Editor Comments (if provided):

Authors should address reviewer questions and comments.

In addition, authors should note that their Figures 4-7 demonstrate that accidental parathyroid removal & hypo-parathyroidism is worse with CN (experimental) than with control. This is the opposite of what the authors report in the results and discussion and conclusion. Please explain. it is either a mistake in conclusion or a mistake in the way the treatment assignment variable direction is coded.

Also, authors should report the chance-corrected agreement (i.e. kappa) in the screening stages and the eventual final set of included studies.

Reviewers' comments:

Reviewer's Responses to Questions

**Comments to the Author**

1. Is the manuscript technically sound, and do the data support the conclusions?

Reviewer #1: Partly

2. Has the statistical analysis been performed appropriately and rigorously? 

Reviewer #1: I Don't Know

3. Have the authors made all data underlying the findings in their manuscript fully available?

Reviewer #1: Yes

4. Is the manuscript presented in an intelligible fashion and written in standard English?

Reviewer #1: No

5. Review Comments to the Author

Reviewer #1: The authors have performed a systematic review with meta-analysis of the use of carbon nanoparticles in thyroid surgery. There are several areas where this manuscript could be improved for clarity throughout. I would suggest re-review for English language. There are several grammatical errors and typos within the manuscript. Additionally, much of the methods and results are not clear in the manuscript and require additional description in the manuscript. With additional clarity of these issues, this could be an interesting paper.

Methods

1. Please clarity the diagnosis included in the paper. Thyroid carcinoma is a varied and diverse group of diseases, ranging from anaplastic to papillary, which in the recent ATA guidelines has been downgraded in many cases. It needs to be clear which disease you are examining in the review.

2. Why did you only include studies that include single medical team/surgeon? This seems like an odd inclusion criteria.

3. For your exclusion criteria, what did you include for "incomplete data"? You provide to description of what is considered complete data in your inclusion criteria. Do patient need to have follow up? Data on calcium levels? Of note, many of the included papers do not have simple data, like the average age. Why were these papers included while others were not?

4. In the abstract is it stated that only RCTs or non randomized CT are included however in the methods, it appears that retrospective trials were included. Can you please clarify this contradiction?

Results

1. Please note the results in the figure diagrams are unclear. I cannot confirm the number of studies included or excluded and for which reasons based on the formatting errors.

2. The actual patient population and surgeries performed in the respective studies in unclear - how many were reoperations? How many were original surgeries? This is important data for the reader to be able to see and assess

3. Examining the results and figure, there is a high degree of variability in the actual number of studies included. I can't seem to find a single figure that includes all 25 included papers. Many of the sub analyses have small sample sizes (2 papers), which is quite different that the 25 papers reported in the results. Each sub analysis should be well described in the methods and results section, with a clearly states number or papers included and sample size for that analysis. Without this, the results of your paper are presented in a misleading way, implying there is more power to you analysis than their actual is

Figures and tables

1. I would suggest editing these figures and tables for consistent text format and readability. The content is cut off in several of the images. Additionally, there are far too many figures and tables, with scattered data throughout that is not easier for the reader to assess and interpret for themselves.

6. PLOS authors have the option to publish the peer review history of their article (what does this mean?). If published, this will include your full peer review and any attached files.

Reviewer #1: No

---

## [Author Response · Author response to Decision Letter 0]

15 Jan 2020

Dear reviewers, 

Thank you for your review of our manuscript: PONE-D-19-26418 - “The Role of Carbon Nanoparticle in Lymph Node Detection and Parathyroid Gland Protection during Thyroidectomy- a Meta Analysis”. We really appreciate the reviewers’ valuable comments. We are resubmitting a revised manuscript, per their suggestions. We describe below the changes we have made as well as our response to the reviewer’s comments. The main changes were highlighted in red.

Journal Requirements:

1. When submitting your revision, we need you to address these additional requirements. Please ensure that your manuscript meets PLOS ONE's style requirements, including those for file naming.

Response: 

The manuscript has been revised according to PLOS ONE's style requirements.

2. Please provide the full search strategy and full search terms used for at least one database used in the supplementary information. In addition, please provide additional details regarding the assessment of publication bias in your systematic review and meta-analysis. Specifically, please provide the funnel plots calculated from the Begg's test and/or the regression plots from the Egger's test.

Response: 

The full search strategy and full search terms used has been listed in the Method section: page 4, line 69-72 . In addition, the funnel plots calculated from the Begg's test has been added in the text: page 7, line 125-126.

Response: 

Yes, I will provide the relevant information if needed.

Reviewers’ comments: 

1. In addition, authors should note that their Figures 4-7 demonstrate that accidental parathyroid removal & hypo-parathyroidism is worse with CN (experimental) than with control. This is the opposite of what the authors report in the results and discussion and conclusion. Please explain. it is either a mistake in conclusion or a mistake in the way the treatment assignment variable direction is coded.

Response: 

Thanks for your careful review. I made a mistake in the way the treatment assignment variable direction is coded, which have been corrected in the revised manuscript.(Modified at line 149, line 154, and line 163)

2. Also, authors should report the chance-corrected agreement (i.e. kappa) in the screening stages and the eventual final set of included studies.

Response: 

The chance-corrected agreement has been added and inserted in page 6, line 117.

3. Please clarity the diagnosis included in the paper. Thyroid carcinoma is a varied and diverse group of diseases, ranging from anaplastic to papillary, which in the recent ATA guidelines has been downgraded in many cases. It needs to be clear which disease you are examining in the review.

Response: 

Papillary thyroid carcinoma, follicular thyroid carcinoma, and medullary thyroid carcinoma comprising the subjects of the 25 studies included in the current meta-analysis. It’s impossible to distinguish them base on the data available. I combined these 3 pathologic types and termed as Non-anaplastic thyroid carcinoma, and make revision in the text where appropriate. The details were inserted in page 6, line 119-121.

4. Why did you only include studies that include single medical team/surgeon? This seems like an odd inclusion criteria.

Response: 

I delete the criteria in manuscript. ( Modified in page 4, line 78-79.)

5. For your exclusion criteria, what did you include for "incomplete data"? You provide to description of what is considered complete data in your inclusion criteria. Do patients need to have follow up? Data on calcium levels? Of note, many of the included papers do not have simple data, like the average age. Why were these papers included while others were not?

Response: 

Thanks for your critical review. “Incomplete data” is vague and inaccurate, so I deleted this exclusion criteria. (Modified in page 4, line 82-83.)

In addition, only 7 studies had a follow-up time of at least 6 months and they are eligible for the subgroup analysis of postoperative permanent hypocalcemia/hypoparathyroidism. I mentioned this in page 14, line 173. 

6. In the abstract is it stated that only RCTs or non randomized CT are included however in the methods, it appears that retrospective trials were included. Can you please clarify this contradiction?

Response: 

All the studies included in the meta-analsis were RCTs or Non-RCTs, and there was no retrospective trials included. I have corrected in the appropriate part of the text: page 5, line 98-101. 

7. Please note the results in the figure diagrams are unclear. I cannot confirm the number of studies included or excluded and for which reasons based on the formatting errors.

Response: 

The figures have been corrected and inserted in the appropriate place in the text.

8. The actual patient population and surgeries performed in the respective studies is unclear - how many were reoperations? How many were original surgeries? This is important data for the reader to be able to see and assess

Response:

I added the mentioned data in page 6, line 122-123.

9. Examining the results and figure, there is a high degree of variability in the actual number of studies included. I can't seem to find a single figure that includes all 25 included papers. Many of the sub analyses have small sample sizes (2 papers), which is quite different that the 25 papers reported in the results. Each sub analysis should be well described in the methods and results section, with a clearly states number or papers included and sample size for that analysis. Without this, the results of your paper are presented in a misleading way, implying there is more power to your analysis than their actual is

Response:

Thanks for your comment. In fact, studies included had different objectives and provided only their associate data. Therefore, there was no analysis including 25 studies. In addition, I add the data of number or papers included and sample size for each subgroup analysis according to your advice. (Modified at 112-185 )

10. I would suggest editing these figures and tables for consistent text format and readability. The content is cut off in several of the images. Additionally, there are far too many figures and tables, with scattered data throughout that is not easier for the reader to assess and interpret for themselves.

Answer: Thank you for your kindly advice. I summarized the result of each subgroup analysis in Table 2 for more intuitive.

We look forward to your acceptance.

Best regards,

Peng Hanwei.

---

## [Editor Report · Decision Letter 1]

15 May 2020

PONE-D-19-26418R1

The Role of Carbon Nanoparticle in Lymph Node Detection and Parathyroid Gland Protection during Thyroidectomy for Non-Anaplastic Thyroid Carcinoma- a Meta-Analysis

PLOS ONE

Dear Prof. Peng,

Thank you for submitting your manuscript to PLOS ONE. After careful consideration, we feel that it has merit but does not fully meet PLOS ONE’s publication criteria as it currently stands. Therefore, we invite you to submit a revised version of the manuscript that addresses the points raised during the review process.

We would appreciate receiving your revised manuscript by Jun 29 2020 11:59PM. To enhance the reproducibility of your results, we recommend that if applicable you deposit your laboratory protocols in protocols.io, where a protocol can be assigned its own identifier (DOI) such that it can be cited independently in the future. For instructions see: http://journals.plos.org/plosone/s/submission-guidelines#loc-laboratory-protocols

We look forward to receiving your revised manuscript.

Kind regards,

Biniam Kidane

Academic Editor

PLOS ONE

Additional Editor Comments (if provided):

-Fig 1 is cut off. It needs to be fixed please.

-line 141-142 and Fig 3 are discordant: "WMD analysis showed that the total number of harvested lymph nodes in the CN group was significantly higher than that in the control groups" however Fig 3 shows the opposite. Please address this.

-Supplemental figures 1 and 2 also show the opposite of your conclusions; they show that control performs better whereas your conclusions state that experimental (CN) performs better.

-Supplemental figure 3 and line 179-183: this is inappropriate use of fixed effects modeling. With only 2 studies dealing with heterogenous populations and also including comparisons with near-zero values, there is an unacceptably high risk of unstable estimates with fixed effects modeling. This should be done with random effects.

---

## [Author Response · Author response to Decision Letter 1]

14 Jun 2020

Dear reviewers, 

Thank you for your review of our manuscript: PONE-D-19-26418 - “The Role of Carbon Nanoparticle in Lymph Node Detection and Parathyroid Gland Protection during Thyroidectomy- a Meta Analysis”. We really appreciate the reviewers’ valuable comments. We are resubmitting a revised manuscript, per their suggestions. We describe below the changes we have made as well as our responses to the reviewer’s comments. The main changes were highlighted in red.

1. Fig 1 is cut off. It needs to be fixed please.

Response:

We have fixed the Fig 1 in the manuscript. (Modified at line 124-125 )

2. Line 141-142 and Fig 3 are discordant: "WMD analysis showed that the total number of harvested lymph nodes in the CN group was significantly higher than that in the control groups" however Fig 3 shows the opposite. Please address this.

Response：

Thanks for your sharp review. We made a mistake in the way the treatment assignment variable direction is coded, which have been corrected in the revised manuscript. (Fig3, Modified at line 147)

3. Supplemental figures 1 and 2 also show the opposite of your conclusions; they show that control performs better whereas your conclusions state that experimental (CN) performs better.

Response：

We are so sorry that we made the same mistakes in our first revision in the Supplemental fig 1 and fig 2. We have revised these data and check throughout the manuscript to confirm that there is no similar mistakes remained.

4. Supplemental figure 3 and line 179-183: this is inappropriate use of fixed effects modeling. With only 2 studies dealing with heterogenous populations and also including comparisons with near-zero values, there is an unacceptably high risk of unstable estimates with fixed effects modeling. This should be done with random effects.

Response：

Thank you for your kindly advise. Random effects have been used in Supplemental figure 3, figure 4, figure5 and the related text in the manuscript have been revised. (See line 32-33, 170, 173-174, 177-180, Table2)

Thank you for your consideration of our revised manuscript. We look forward to the re-review of our manuscript and are prepared to make additional revisions as required.

Yours Faithfully,

Hanwei Peng

---

## [Decision Letter · Decision Letter 2]

20 Aug 2020

PONE-D-19-26418R2

The Role of Carbon Nanoparticle in Lymph Node Detection and Parathyroid Gland Protection during Thyroidectomy for Non-Anaplastic Thyroid Carcinoma- a Meta-Analysis

PLOS ONE

Dear Dr. Peng,

Thank you for submitting your manuscript to PLOS ONE. After careful consideration, we feel that it has merit but does not fully meet PLOS ONE’s publication criteria as it currently stands. Therefore, we invite you to submit a revised version of the manuscript that addresses the points raised during the review process.

There are still some issues raised by the reviewers and some additional edits that would aid the manuscript, especially improving the description of the CN technique in a more detailed manner.

We look forward to receiving your revised manuscript.

Kind regards,

Claudio Andaloro

Academic Editor

PLOS ONE

Reviewers' comments:

Reviewer's Responses to Questions

**Comments to the Author**

1. If the authors have adequately addressed your comments raised in a previous round of review and you feel that this manuscript is now acceptable for publication, you may indicate that here to bypass the “Comments to the Author” section, enter your conflict of interest statement in the “Confidential to Editor” section, and submit your "Accept" recommendation.

Reviewer #2: (No Response)

Reviewer #3: (No Response)

2. Is the manuscript technically sound, and do the data support the conclusions?

Reviewer #2: Yes

Reviewer #3: Yes

3. Has the statistical analysis been performed appropriately and rigorously? 

Reviewer #2: I Don't Know

Reviewer #3: I Don't Know

4. Have the authors made all data underlying the findings in their manuscript fully available?

Reviewer #2: Yes

Reviewer #3: Yes

5. Is the manuscript presented in an intelligible fashion and written in standard English?

Reviewer #2: No

Reviewer #3: Yes

6. Review Comments to the Author

Reviewer #2: Major comments:

How exactly does using CN help identify parathyroid glands? Is it just that the parathyroid glands are not mistaken for lymph nodes since the lymph nodes would turn black? Are parathyroid glands commonly excised because they are thought to be lymph nodes or is it simply because they get stuck to the capsule? Please clarify specifically how the CN would help identify parathyroid glands.

If the CN helps identify more lymph nodes but does NOT help identify more diseased lymph nodes, is it worth utilizing? What would be the benefit of identifying and excising more benign lymph nodes?

Minor comments:

Please include the pseudo confidence interval as part of the legend of Figure 2

There are a number of punctuation and grammatical errors throughout the manuscript, please review carefully and correct these

Reviewer #3: Introduction: Please clear the matter of the high percentages that you described as post surgical incidence of hypocalcemia and hypoparathyroidm: they are valid only in transient cases. Permanent hypocalcemia or hypoparathyroidism are described in the literature usually in less than 2% of cases.

Please describe the CN technique more detailed.

Material: Please explain the reason for with you did consider only adolescents over 16 yers old and adults in your study.

Please define more clearly " Included patients with benign thyroid disease unable to be separated from N-ATC.

Results: Please describe the definition of transient respective;;y permanent hypocalcamie/hypoparathyroidism. Some studies use the PTH value less than 10 pg/mL, others less than 7 pg/mL. Some do state the permanent condition six months after the surgery, other use a different time interval. Please do comments which threshold and time interval were considered in the evaluated studies.

Discussions: Till this section there was do specific talk about transient versus permanent hypoparathyroidism. Also the number pf references used, when comparing the current results with the previous results is small and insuficient.

7. PLOS authors have the option to publish the peer review history of their article (what does this mean?). If published, this will include your full peer review and any attached files.

Reviewer #2: No

Reviewer #3: No

---

## [Author Response · Author response to Decision Letter 2]

15 Sep 2020

Dear reviewers, 

Thank you for your review of our manuscript: PONE-D-19-26418 - “The Role of Carbon Nanoparticle in Lymph Node Detection and Parathyroid Gland Protection during Thyroidectomy- a Meta Analysis”. We really appreciate the reviewers’ valuable comments. We are resubmitting a revised manuscript, per their suggestions. We describe below the changes we have made as well as our responses to the reviewer’s comments. The main changes were highlighted in red.

Q1: How exactly does using CN help identify parathyroid glands? Is it just that the parathyroid glands are not mistaken for lymph nodes since the lymph nodes would turn black? Are parathyroid glands commonly excised because they are thought to be lymph nodes or is it simply because they get stuck to the capsule? Please clarify specifically how the CN would help identify parathyroid glands.

Response: CN technique was described in detail in introdunction. (Modified at line 54-59)

Q2: Please include the pseudo confidence interval as part of the legend of Figure 2

Response: We describe the 95% confidence interval in results. (Modified at line 127)

Q3: There are a number of punctuation and grammatical errors throughout the manuscript, please review carefully and correct these

Response: Thanks for your sharp review. We made a mistake in punctuation and grammatical errors, which have been corrected in the revised manuscript. (Modified at line 108, 115-117, 138, 153, 160, 174, 184, 254)

Q4: Please clear the matter of the high percentages that you described as post surgical incidence of hypocalcemia and hypoparathyroidism: they are valid only in transient cases. Permanent hypocalcemia or hypoparathyroidism are described in the literature usually in less than 2% of cases.

Response: A more accurate expression was modified as your kindly advise. (Modified at line 46-47)

Q5: Please describe the CN technique more detailed. 

Response: The same as Q1.

Q6: Please explain the reason for with you did consider only adolescents over 16 years old and adults in your study.

Response: To reduce the publication bias because most of the studies included only report the cases in adult or adolescents over 16 years old. (Modified at line 87-89)

Q7: Please define more clearly " Included patients with benign thyroid disease unable to be separated from N-ATC.

Response: Thank you for your kindly advise. We have modified it more clearly in exclusion criteria. (Modified at line 86-87)

Q8: Please describe the definition of transient respectively permanent hypocalcamie/hypoparathyroidism. Some studies use the PTH value less than 10 pg/mL, others less than 7 pg/mL. Some do state the permanent condition six months after the surgery, other use a different time interval. Please do comments which threshold and time interval were considered in the evaluated studies.

Response: We describe the definition of hypoparathyroidism and hypocalcemia in Discussion. (Modified at line 224-228)

Actually, most studies included in meta-analysis report the cases of hypoparathyroidism and hypocalcemia, but not the value of PTH in detail.

Thank you for your consideration of our revised manuscript. We look forward to the re-review of our manuscript and are prepared to make additional revisions as required.

Yours Faithfully,

Hanwei Peng

---

## [Decision Letter · Decision Letter 3]

30 Sep 2020

PONE-D-19-26418R3

The Role of Carbon Nanoparticle in Lymph Node Detection and Parathyroid Gland Protection during Thyroidectomy for Non-Anaplastic Thyroid Carcinoma- a Meta-Analysis

PLOS ONE

Dear Dr. Peng,

Thank you for submitting your manuscript to PLOS ONE. After careful consideration, we feel that it has merit but does not fully meet PLOS ONE’s publication criteria as it currently stands. Therefore, we invite you to submit a revised version of the manuscript that addresses the points raised during the review process.

Dear Authors, You did not answer some of previous points raised by reviewer. Please pay attention to this matter. Moreover, a reviewer raised the following question "If the CN helps identify more lymph nodes but does NOT help identify more diseased lymph nodes, is it worth utilizing?  What would be the benefit of identifying and excising more benign lymph nodes?" It requires an explanation

the question about hypoparathyroidism definition is still not addressed,

We look forward to receiving your revised manuscript.

Kind regards,

Claudio Andaloro

Academic Editor

PLOS ONE

Reviewers' comments:

Reviewer's Responses to Questions

**Comments to the Author**

1. If the authors have adequately addressed your comments raised in a previous round of review and you feel that this manuscript is now acceptable for publication, you may indicate that here to bypass the “Comments to the Author” section, enter your conflict of interest statement in the “Confidential to Editor” section, and submit your "Accept" recommendation.

Reviewer #3: (No Response)

2. Is the manuscript technically sound, and do the data support the conclusions?

Reviewer #3: Yes

3. Has the statistical analysis been performed appropriately and rigorously? 

Reviewer #3: Yes

4. Have the authors made all data underlying the findings in their manuscript fully available?

Reviewer #3: Yes

5. Is the manuscript presented in an intelligible fashion and written in standard English?

Reviewer #3: Yes

6. Review Comments to the Author

Reviewer #3: The question about hypoparathyroidism definition is still not addressed. Row 224-228 from discussions o not contain informations about hypoparathyroidism and hypocalcemia - transient versus permanent. This does not play a major role in the manuscript results but does really influence the motivation and the importance of the technique described in the study.

7. PLOS authors have the option to publish the peer review history of their article (what does this mean?). If published, this will include your full peer review and any attached files.

Reviewer #3: No

---

## [Author Response · Author response to Decision Letter 3]

3 Oct 2020

Dear reviewers, 

Thank you for your prompt re-review of our manuscript: PONE-D-19-26418 - “The Role of Carbon Nanoparticle in Lymph Node Detection and Parathyroid Gland Protection during Thyroidectomy- a Meta Analysis”. After discussing the comments raised by the reviewer, we made changes per the reviewers’ comments. We are now resubmitting a revised manuscript to correct the errors together with this cover letter for further evaluation. 

Q1: Hypoparathyroidism definition is still not addressed. 

Response: We addressed the hypoparathyroidism definition as well as diagnostic criteria in the current study in the manuscript. (See line 104-110)

Q2: If the CN helps identify more lymph nodes but does NOT help identify more diseased lymph nodes, is it worth utilizing? What would be the benefit of identifying and excising more benign lymph nodes? 

Response: Thanks for your question. This is really a crucial point that readers may concern. In fact, in the results section, we stated " WMD analysis showed that the total number of harvested lymph nodes in the CN group was significantly higher than that in the control groups (WMD, 2.36; 95% CI, 1.40 to 3.32; P＜0.01, Fig 3).” (see line 157-159) and “No difference was found between the CNs group and the Control group regarding LN metastatic rate (OR = 1.07, 95% CI = 0.75 to 1.51, P=0.71, Fig 4).” (See line 163-164). These denoted that the numbers of both diseased lymph nodes and benign lymph nodes were higher in the CN group than that in the control group. Therefore, we advocate that this technique is worth utilizing. We revised a sentence in the discussion section (line 229-231) and added a sentence in this paragraph (line 234-237) to express our points of view. 

Thank you for your consideration of our re-revised manuscript. We look forward to further review of our manuscript and are prepared to make additional revisions as required.

Yours Faithfully,

Hanwei Peng

---

## [Editor Report · Decision Letter 4]

7 Oct 2020

The Role of Carbon Nanoparticle in Lymph Node Detection and Parathyroid Gland Protection during Thyroidectomy for Non-Anaplastic Thyroid Carcinoma- a Meta-Analysis

PONE-D-19-26418R4

Dear Dr. Peng,

We’re pleased to inform you that your manuscript has been judged scientifically suitable for publication and will be formally accepted for publication once it meets all outstanding technical requirements.

Kind regards,

Claudio Andaloro

Academic Editor

PLOS ONE
---

## [Editor Report · Acceptance letter]

15 Oct 2020

PONE-D-19-26418R4 

The Role of Carbon Nanoparticle in Lymph Node Detection and Parathyroid Gland Protection during Thyroidectomy for Non-Anaplastic Thyroid Carcinoma- a Meta-Analysis 

Dear Dr. Peng:

I'm pleased to inform you that your manuscript has been deemed suitable for publication in PLOS ONE. Congratulations! Your manuscript is now with our production department. 

Kind regards, 

on behalf of

Dr. Claudio Andaloro 

Academic Editor

PLOS ONE